# Socio-Demographic Predictors of Food Waste Behavior in Denmark and Spain

**Alessandra C. Grasso** [1,2,*] **, Margreet R. Olthof** [1,2]**, Anja J. Boevé** [1,2]**, Corné van Dooren** [3]**, Liisa Lähteenmäki** [4] **and Ingeborg A. Brouwer** [1,2]

1    Department of Health Sciences, Faculty of Science, Vrije Universiteit Amsterdam, 1081 HV Amsterdam, The Netherlands; margreet.olthof@vu.nl (M.R.O.); a.j.boeve@vu.nl (A.J.B.); ingeborg.brouwer@vu.nl (I.A.B.)
2    Amsterdam Public Health Research Institute, 1081 BT Amsterdam, The Netherlands
3    Netherlands Nutrition Centre (Voedingscentrum), 2594 AC The Hague, The Netherlands; dooren@voedingscentrum.nl
4    MAPP Centre, Aarhus University, DK-8210 Aarhus V, Denmark; liisal@mgmt.au.dk
*    Correspondence: alessandra.grasso@vu.nl; Tel.: +31-20-59-84038

**Abstract:** Food waste generated at the household level represents about half of the total food waste in high-income countries, making consumers a target for food waste reduction strategies. To successfully reduce consumer food waste, it is necessary to have an understanding of factors influencing food waste behaviors (FWB). The objective of this study was to investigate socio-demographic predictors of FWB among consumers in two European countries: Denmark and Spain. Based on a survey involving 1518 Danish and 1511 Spanish consumers, we examined the associations of age, sex, education, marital status, employment status, and household size with FWB. By using structural equation modeling based on confirmatory factor analysis, we created the variable FWB from self-reported food waste and two activities that have been correlated with the amount of food wasted in previous studies: namely, shopping routines and food preparation. Results show that being older, unemployed, and working part-time were associated with less food waste behavior in both countries. In Denmark, being male was associated with more food waste behavior, and living in a household with four or more people was associated with less food waste behavior. These results underscore the modest role of socio-demographic characteristics in predicting food waste behavior in Europe.

**Keywords:** food waste; behavior; socio-demographic; predictors; SEM

## 1. Introduction

One-third of food produced for human consumption—approximately 1.3 billion tons per year—gets lost or wasted globally [1]. This amounts to a considerable waste of the resources that are used in food production, such as cropland, water, energy, and fertilizers, as well as the large amounts of greenhouse gas emissions associated with food production and food waste disposal [2]. Not only does food waste lead to unnecessary and avoidable environmental degradation such as soil erosion and biodiversity loss, it induces economic losses and perpetuates social inequalities due to lower wages across the food system, higher food prices, and the widening of the food access gap [3,4]. Food waste occurs at all points throughout the food supply chain, but in Europe, the majority (53%) of the total food waste is generated at the household level, totaling about 47 million tons of food waste in 2012 [5]. As households, and thus consumers, produce the most food waste, it is important to know which factors influence consumer-level food waste to tackle the aforementioned environmental, economic, and social issues.

Consumer-related food waste is a complex and multi-faceted issue that is influenced by cultural, social, political, economic, and geographic drivers, as well as cognitive, motivational, and structural factors, food-related behaviors, and food habits [6,7]. The theory of planned behavior [8] is the more frequently applied theoretical framework that has been used to explain or predict consumer-level food waste [9–13]. The theory suggests that behavior is determined by intention, which is influenced by subjective norm, perceived behavioral control, and attitude [8]. Yet, intention does not correspond well with behavior in all cases. Several studies have demonstrated that food-related behaviors such as planning and shopping routines, when added or compared to the theory of planned behavior, are more important indicators than intention for the amount of food wasted [10,11].

Furthermore, a sociological approach to food waste has highlighted the importance of the social and material contexts of everyday food waste practices, pointing out that factors such as time, domestic divisions of labor with regard to food shopping and preparation, and infrastructures of provision influence food waste behavior (FWB), but may be beyond the control of consumers [14,15]. Consequently, factors related to socio-demographics such as sex and employment status may influence food provisioning practices, leading to food waste [16]. However, evidence suggests that socio-demographic characteristics have weak predictive power of food waste [10,17]. Previous studies have found socio-demographics to explain only 7–13% of the variance regarding intention to reduce and perceived behavioral control to avoid household food waste [12,18]. Despite the limited predictive power, correlations between age, sex, employment status, income, household size and composition, and amount of food wastes have been found, but the strength and direction of the relationships vary between studies [7]. In addition, such studies have examined the correlations between socio-demographic characteristics and self-reported food waste, but have not looked at these alongside food-related behaviors that are known to influence food waste.

Self-reported food waste is a major limitation in most studies, as it may suffer from social desirability and hypothetical bias, and consequently, may deviate from actual behavior [19,20]. However, more objective techniques to measure food waste, such as waste composition analysis or diary-based methods, are timely and financially costly [20,21]. Thus, some researchers have focused on food waste as an aggregate of food-related behaviors rather than on self-reported food waste as an outcome. For instance, Mondejar-Jiménez et al.'s outcome regarding 'positive behavior toward food waste' consisted of food-related activities that have been found to influence the amount of food wasted at the consumer level, namely reusing leftovers, understanding the date labels on foods, and making a shopping list [13]. As there is limited literature using such an outcome to explore factors influencing food waste behavior, further investigation into its application is warranted.

The objective of this study was to investigate socio-demographic predictors of FWB among consumers in two European countries, namely Denmark and Spain. In Denmark, it is estimated that the average household wastes about 183 kg food per year [22], while in Spain, it is estimated that the average household wastes about 71.2 kg food per year [23]. By employing a similar approach to Mondejar-Jiménez et al., this study tries to move beyond using self-reported food waste as the exclusive measure of this behavior. The results presented in the paper can add to the existing literature on the role that socio-demographics play in predicting food waste behavior.

## 2. Materials and Methods

### 2.1. Procedure and Sample

Data collection was carried out with Qualtrics—a panel service agency—in June and July 2014. An online questionnaire was made available to a randomly selected sample of panelists in Denmark and Spain. The questionnaire was developed in English, translated to Danish and Spanish, and distributed through online platforms to Danish and Spanish respondents, respectively. Qualtrics follows the European Society for Opinion and Marketing Research (ESOMAR) principles in their data collection activities and panel management. The respondents had to confirm their willingness to

participate in the study, and their data was handled with anonymity and confidentiality in accordance with the provisions of the Declaration of Helsinki [24].

In total, 3034 respondents completed the questionnaire, with 1522 respondents from Denmark and 1512 respondents from Spain. Of the 3034 respondents who completed the questionnaire, five had missing data for age, and were excluded from this study. Thus, 1518 Danish respondents and 1511 Spanish respondents had complete details for all the relevant variables. Table 1 provides a summary of the socio-demographic characteristics of the respondents included in this study. The majority of the respondents were responsible to some extent for the provision of food in their household: 76% in both countries were responsible for deciding what food to cook/prepare for household meals, 83% in Denmark and 90% in Spain were responsible for food shopping, and 76% in Denmark and 79% in Spain were responsible for cooking and preparing food.

**Table 1.** Socio-demographic characteristics of sample in Denmark (N = 1518) and Spain (N = 1511).

| Characteristic | Denmark | Spain |
|---|---|---|
| Age (years), mean | 50.1 | 37.0 |
| Sex | | |
| Female | 48.4% | 48.8% |
| Male | 51.6% | 51.2% |
| Education [1] | | |
| Low | 24.1 % | 11.3% |
| Middle | 40.8 % | 34.4% |
| High | 35.1 % | 54.3% |
| Employment status [2] | | |
| Full time | 42.0% | 53.1% |
| Part time | 11.1% | 14.8% |
| Unemployed | 13.6 % | 32.2% |
| Retired | 33.2 % | Not asked |
| Marital status [3] | | |
| Married/living with partner | 59.5% | 60.3% |
| Single | 40.5% | 39.7% |
| Household size (number of persons) | | |
| 1 | 29.1% | 7.9% |
| 2 | 43.8% | 23.5% |
| 3 | 11.4% | 30.3% |
| 4 | 10.5% | 27.9% |
| 5+ | 5.1% | 10.3% |

[1] Low includes lower primary to lower secondary school, middle includes upper secondary school to additional training; higher includes Bachelor or other higher education. [2] Full time if working a minimum of 30 hours per week, part-time if working between 15 and29 hours per week. [3] Single includes widowed, divorced and separated. [6] Categories are married or separated; never married, widowed or divorced.

## 2.2. Measures

The questionnaire contained measures on food-related behaviors, self-reported food waste, and socio-demographics. The study was a part of a larger survey on food-related behavior and mental well-being, which has been described elsewhere [25]. This study reports the findings related to food-related behaviors that are likely to increase food waste and how they are linked to socio-demographic characteristics. The used measures, described below, were selected based on earlier findings of behaviors that have been linked to reported food waste, and are reported in Table 2.

**Table 2.** Latent variables and indicators in the hypothesized measurement model of food waste behavior.

| Latent Variables (LV) | Indicators (Q) |
|---|---|
| LV1: Planning routines | Qp1. The shopping trips are usually planned in advance (shopping lists are made, inventories are checked, etc.). [1]<br>Qp2.The home meals are usually planned a couple of days ahead. [1] |
| LV2: Shopping routines | Qs1. In general, I buy too much food when shopping (e.g. more than I end up using).<br>Qs2. I often buy unintended food products when shopping.<br>Qs3. I often buy food in packages that are too big for my needs.<br>Qs4. I usually buy higher amounts of food when they offer good value for money. |
| LV3: Leftover management | Ql1. I always throw out products that are beyond the best-before date.<br>Ql2. I always reuse leftovers. [1] |
| LV4: Self-reported food waste | How much … would you say that you throw away, of what you buy and/or grow, in a regular week?<br>Qfw1. Food<br>Qfw2. Milk and other dairy products<br>Qfw3. Fresh fruits and vegetables<br>Qfw4. Meat and fish<br>Qfw5. Bread and other bakery products |
| LV5: Food waste behavior (FWB) | LV1. Planning routines<br>LV2. Shopping routines<br>LV3. Food leftover management<br>LV4. Self-reported food waste<br>Qc1. Food preparation: Too much food is often cooked/prepared for a meal. |

[1] Scale was reversed for analyses.

### 2.2.1. Food-Related Behaviors and Self-Reported Food Waste

The food-related behaviors that were assessed included planning routines, shopping routines, practices related to handling leftovers and food beyond its best-before date, and food preparation (Table 2). Planning routines consisted of two items, referring to the planning of shopping trips and meals [10]. Shopping routines were measured with four items concerning the excess purchase of food [26]. Leftover management was measured with two items about reusing leftovers and throwing out products that are beyond the best-before date. Food preparation was assessed with one item concerning preparing/cooking more food than needed. These variables were measured by having respondents rate their agreement to statements using a 7-point Likert scale in the Danish questionnaire and a 5-point Likert scale in the Spanish questionnaire.

Self-reported food waste consisted of five items, referring to food waste in general and four specific food sub-categories, namely milk and other dairy products, fresh fruits and vegetables, meat and fish, and bread and other bakery products [11], and was measured by having respondents report how often they throw away the food in a regular week (not at all; less than 1/10th; more than 1/10th but less than 1/4; more than 1/4 but less than 1/2; more than 1/2).

### 2.2.2. Socio-Demographics

Respondents were asked to indicate their age, sex, education, employment status, marital status (married or living with partner; single; widowed; divorced; or separated), and household size (1; 2; 3; 4; 5; or 6+). The following categories of education were created: low includes lower primary to lower secondary school, middle includes upper secondary school to additional training; and higher includes Bachelor or other higher education. Full-time employment was defined in the questionnaire as a minimum of 30 hours work per week, while part-time employment was defined as between 15 and 29 hours work per week. In addition, those who reported to be widowed, divorced, or separated were considered as single in this study. Since only a few people reported to live in a household with

6+ people (19 in Denmark and 24 in Spain), we grouped them with those who reported to live in a household of 5 people.

### 2.3. Statistical Analysis

All analyses were done separately for each country and were run in RStudio version 1.1.383 (RStudio Inc., Boston, Massachusetts, USA). Descriptive statistics were conducted to report the socio-demographic characteristics of the sample, respondents' agreement toward food-related behaviors, and self-reported food waste in general and by food sub-category.

### 2.3.1. Modeling Food Waste Behavior

Before we were able to examine the socio-demographic predictors of FWB, we needed to model FWB, as it was not directly measured. Similar to Mondejar-Jiménez et al.'s 'positive behavior toward food waste' [13], we created FWB out of self-reported food waste and four food-related activities that have been found to influence the amount of food wasted at the consumer level (Figure 1).

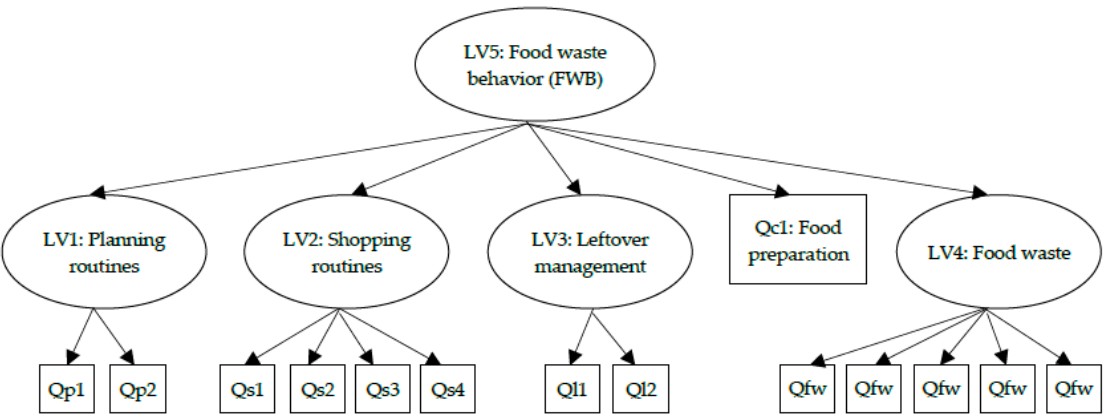

**Figure 1.** Hypothesized measurement model of food waste behavior. Ovals represent latent variables (LV) and rectangles represent indicators (Q) or single questions described in Table 2.

From the food-related behavior perspective, food waste emerges along the chain of events related to household food provisioning, and has been shown to be related to various practices, such as food planning, shopping, storing, cooking, eating, and managing leftovers [14,27]. For instance, making shopping lists [28] and eating leftovers [10] has been associated with less food waste, while buying unintended food products when shopping [10] and throwing out food beyond the best-before date [29] has been associated with more food waste. The food-related activities that we included in our model of FWB are planning routines, shopping routines, food preparation, and practices related to handling leftovers and food beyond its best-before date (Table 2). These activities, which occur at different stages in the chain of events related to household food provisioning, were chosen and combined with self-reported food waste to have a comprehensive measure of FWB.

We modeled FWB using confirmatory factor analysis (CFA). CFA allowed us to test our hypothetical model of FWB as a latent variable (LV) in a simultaneous analysis of the entire system of variables to determine the extent to which it is consistent with the data [30]. The path diagram of the hypothesized measurement model is shown in Figure 1, with a description of the latent variables and indicators in Table 2. The first loading of a LV was set to 1 to give the LV an interpretable scale.

### 2.3.2. Regression Analyses

After testing our hypothesized measurement model of FWB, structural equation modeling (SEM) was conducted to test the latent variable FWB (LV5) as the dependent variable in regression analyses with the following socio-demographic characteristics as independent variables: age, sex, education

level, employment status, marital status, and household size. A backward selection procedure was conducted in which all the socio-demographic predictors of interest were included in the first regression model, and only the statistically significant predictors remained in the final regression model. A *p*-value above the cut-off point of 0.1 was used to remove variables.

## 3. Results

### 3.1. Food-Related Behaviors and Self-Reported Food Waste of Respondents

Figure 2 presents the level of agreement of waste-promoting and waste-reducing (italicized) food-related behaviors by country separately. It appears that the majority of respondents in both countries either somewhat disagree, disagree, or strongly disagree with the statements 'In general, I buy too much food when shopping' and 'I often buy food in packages that are too big for my needs', and somewhat agree, agree, or strongly agree with the statements 'Shopping trips are usually planned in advance', 'I usually buy higher amounts of food when they offer good value for money', and 'I always reuse leftovers'.

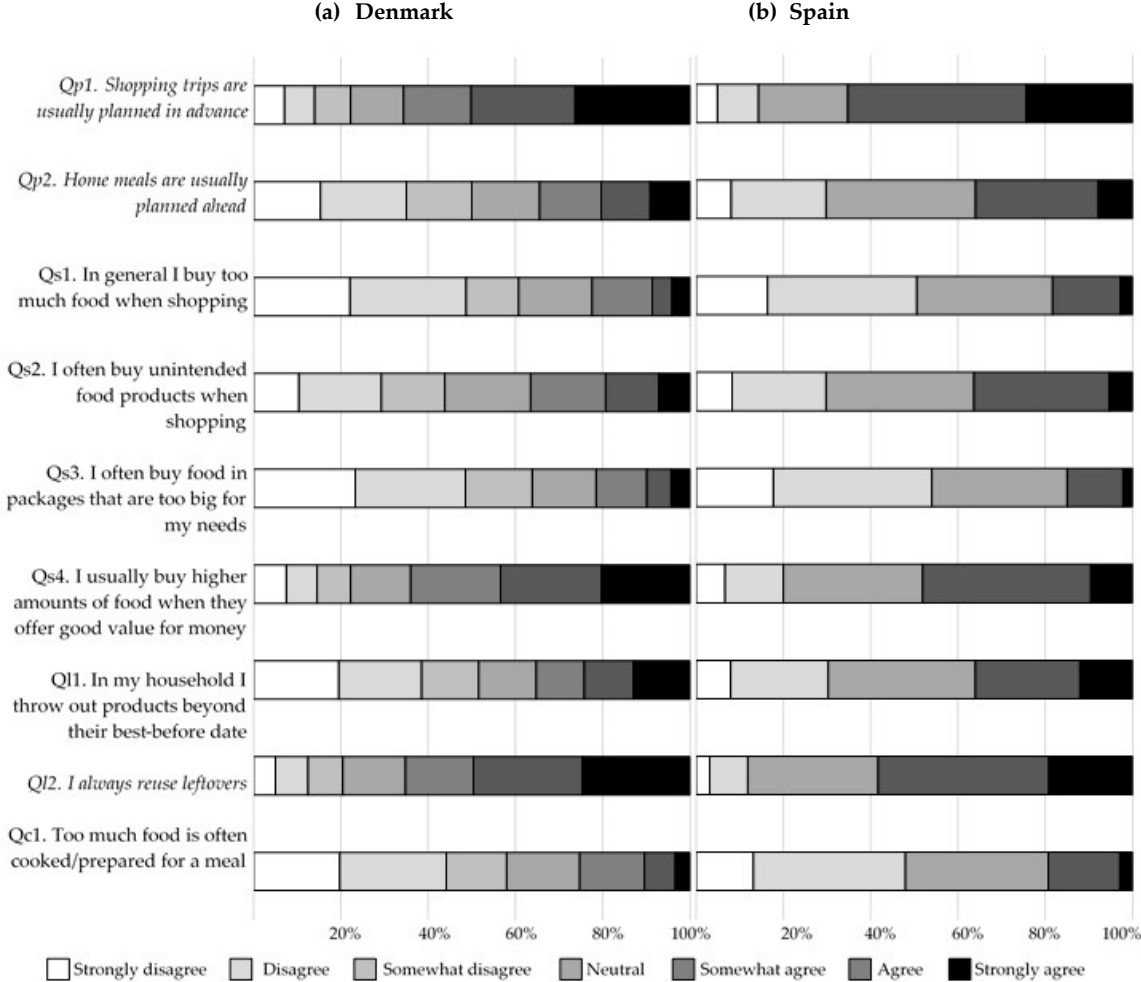

**Figure 2.** Level of agreement of statements on waste-reducing (italicized) and waste-promoting food-related behaviors in respondents in (**a**) Denmark (N = 1518) and (**b**) Spain (N = 1511).

Figure 3 illustrates the amount of food and food sub-categories reported to have been wasted in a regular week for each country separately. Approximately 20% respondents in Denmark and 22% in Spain indicated throwing away more than 10% of food in a regular week, whereas 80% in Denmark and 78% in Spain reported not throwing away any food or less than 10% of food in a regular week.

When looking at the specific food sub-categories, more than 10% of bread and other bakery products were reported to be thrown away by about one-third of respondents in both countries, with fresh fruits and vegetables the second most susceptible food sub-category reported to be thrown away in a regular week in both countries. In Denmark, meat was the least susceptible food sub-category reported to be thrown away in a regular week, while in Spain, milk and other dairy products was the least susceptible sub-category.

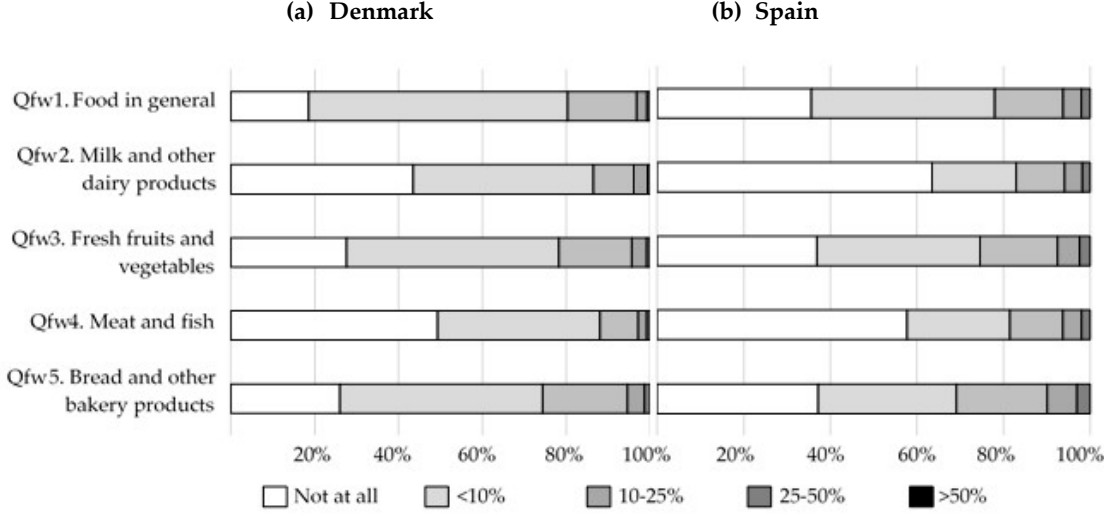

**Figure 3.** Self-reported food waste in Denmark (N = 1518) and Spain (N = 1511) (%). Results of survey question: "How much . . . [see bar for category] would you say that you throw away, of what you buy and/or grow, in a regular week?".

Evaluating the measurement model of FWB revealed that two constructs were problematic: planning routines and leftovers management. The measurement model resulted in negative variance for the first item under planning routines (Qp1) for Denmark, while the measurement model did not converge for Spain. When the measurement of planning routines (LV1) was removed from the measurement model, CFA showed a satisfactory fit of the measurement model for both countries. However, when composite reliability and construct validity were assessed, leftover management (LV3) fell below the cut-off point of 0.7 or greater for CR (0.343 for Denmark and 0.170 for Spain) and 0.50 or greater for AVE (0.207 for Denmark and 0.107 for Spain), and thus was removed from the model. An item (Qs4) under shopping routines (LV2) resulted a poor loading for both countries, but remained in the model because removing it worsened the goodness-of-fit of the model. As illustrated in Figure 4, the resulting measurement model for FWB was satisfactory, as indicated by the CFA overall goodness-of-fit indices (Table 3), with CFI values close to the cut-off value of 0.95 and RMSEA values close to the cut-off value of 0.06 [31]. All the items had significant loadings ($p < 0.001$), and CR and AVE were close to or above their respective cut-off values [33]. The discriminant validity was also satisfactory, with the inter-construct correlation between the remaining first-order LVs (i.e. shopping routines and self-reported food waste) below the threshold of 0.85 (0.445 for Denmark and 0.395 for Spain). Based on the fit indices of the increasingly constrained models used to test for measurement invariance, the chi-square difference tests showed that partial metric invariance holds for the measurement model between Denmark and Spain ($\Delta\chi^2 = 1.176$, d = 1, $p = 0.278$).

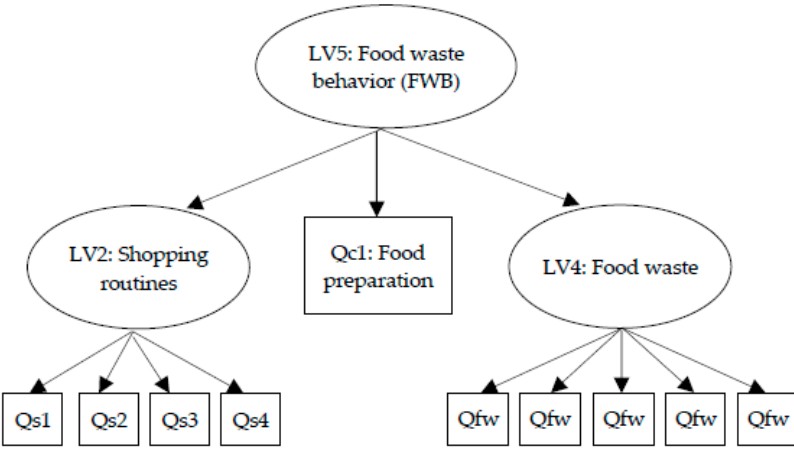

**Figure 4.** Final measurement model of food waste behavior. Ovals represent latent variables (LV) and rectangles represent indicators (Q) or single questions described in Table 3.

**Table 3.** Fit indices and reliability measurements of the final measurement model of food waste behavior for Denmark (DK) and Spain (SP) [1].

| Latent Variables (LV) and Indicators (Q) | Factor Loadings; *CR; AVE* [2] | |
| --- | --- | --- |
| | DK (N = 1518) | SP (N = 1511) |
| LV2: Shopping routines | *0.707; 0.403* | *0.745; 0.430* |
| Qs1. In general, I buy too much food when shopping (e.g., more than I end up using). | 0.847 | 0.792 |
| Qs2. I often buy unintended food products when shopping. | 0.639 | 0.637 |
| Qs3. I often buy food in packages that are too big for my needs. | 0.682 | 0.687 |
| Qs4. I usually buy higher amounts of food when they offer good value for money. | 0.251 | 0.469 |
| LV4: Self-reported food waste | *0.851; 0.534* | *0.918; 0.692* |
| How much … would you say that you throw away, of what you buy and/or grow, in a regular week? | | |
| Qfw1. Food | 0.746 | 0.844 |
| Qfw2. Milk and dairy products | 0.715 | 0.862 |
| Qfw3. Fresh fruits and vegetables | 0.738 | 0.844 |
| Qfw4. Meat and fish | 0.741 | 0.882 |
| Qfw5. Bread and other bakery products | 0.720 | 0.739 |
| LV5. Food waste behavior | *0.737; 0.486* | *0.724; 0.477* |
| LV2. Shopping routines | 0.796 | 0.847 |
| LV4. Self-reported food waste | 0.683 | 0.505 |
| Qc1. Food preparation: Too much food is often cooked/prepared for a meal. | 0.598 | 0.677 |

[1] Goodness of fit indices: Denmark: $\chi^2$ = 144.6, df 33, $p < 0.001$, RMSEA = 0.047, CFI = 0.978; Spain: $\chi^2$ = 285.9, df 33, $p < 0.001$, RMSEA = 0.071, CFI = 0.967. [2] Construct reliability (CR) and average variance extracted (AVE) of each latent variable are italicized.

### 3.2. Prediction Analysis

Table 4 shows the full and final prediction model for Denmark and Spain. Age, sex, employment status, and household size were found to be associated with FWB in Denmark, with 7.3% of the variance of FWB explained by these predictors combined. Being older was associated with less FWB, and being male was associated with more FWB. Compared to those with a full-time job, working part-time or being unemployed or retired were associated with less FWB. Marital status and level of education were not associated with FWB. Compared to living in a single-person household, living in a household with four people was associated with more FWB. The structural equation model for Denmark converged well, and its fit was satisfactory ($\chi^2$ = 505.61, df 114, $p < 0.001$, RMSEA = 0.048, CFI = 0.927).

**Table 4.** Associations of socio-demographic characteristics with food waste behavior from multivariable linear regressions for Denmark and Spain.

| Characteristic | Denmark (N = 1518) | | | | Spain (N = 1511) | | | |
|---|---|---|---|---|---|---|---|---|
| | Full Model | | Final Model | | Full Model | | Final Model | |
| | β [1] | (95% CI) [2] | β | 95% CI | β | 95% CI | β | 95% CI |
| Age (years) | −0.014 *** | (−0.020, −0.008) | −0.014 *** | (−0.020, −0.008) | −0.015 *** | (−0.019, −0.011) | −0.016 *** | (−0.019, −0.012) |
| Sex | | | | | | | | |
| Female (ref) | | | | | | | | |
| Male | 0.151 ** | (0.012, 0.289) | 0.160 ** | (0.022, 0.297) | 0.013 | (−0.069, 0.095) | – | – |
| Education | | | | | | | | |
| Low (ref) | | | | | | | | |
| Middle | −0.104 | (−0.279, 0.071) | – | – | 0.012 | (−0.124, 0.148) | – | – |
| High | −0.049 | (−0.230, 0.132) | – | – | −0.055 | (−0.188, 0.077) | – | – |
| Employment status | | | | | | | | |
| Full-time (ref) | | | | | | | | |
| Part-time | −0.206 * | (−0.440, 0.027) | −0.207 * | (−0.439, 0.025) | −0.096 | (−0.215, 0.024) | −0.077 | (−0.195, 0.041) |
| Not employed | −0.284 ** | (−0.507, −0.060) | −0.278 ** | (−0.497, −0.059) | −0.264 *** | (−0.360, −0.167) | −0.237 *** | (−0.328, −0.145) |
| Retired [3] | −0.219 ** | (−0.409, −0.029) | −0.214 ** | (−0.401, −0.027) | NA | NA | | |
| Marital status | | | | | | | | |
| Married/living with partner (ref) | | | | | | | | |
| Single | −0.016 | (−0.236, 0.205) | – | – | 0.022 | (−0.072, 0.117) | – | – |
| Household size (number of persons) | | | | | | | | |
| 1 (ref) | | | | | | | | |
| 2 | 0.023 | (−0.220, 0.226) | 0.040 | (−0.121, 0.201) | −0.057 | (−0.232, 0.117) | | |
| 3 | 0.039 | (−0.248, 0.325) | 0.050 | (−0.191, 0.290) | 0.020 | (−0.148, 0.187) | | |
| 4 | 0.243 | (−0.068, 0.554) | 0.254 ** | (0.005, 0.503) | 0.111 | (−0.059, 0.280) | | |
| 5+ | −0.159 | (−0.552, 0.203) | −0.147 | (−0.474, 0.181) | 0.151 | (−0.044, 0.347) | – | – |

[1] Beta−coefficient (β). Asterisk if statistically significant * $p < 0.10$, ** $p < 0.05$, *** $p < 0.001$. [2] 95% confidence interval (CI). [3] Not asked (NA) to Spanish respondents.

Age and employment status were found to be associated with FWB in Spain, with 8.1% of the variance of FWB explained by these predictors combined. Being older, working part-time, and being unemployed were associated with less FWB. Marital status, household size, and level of education were not associated with FWB. The structural equation model for Spain converged well, and its fit was satisfactory ($\chi^2$ = 358.98, df 60, $p$ < 0.001, RMSEA = 0.057, CFI = 0.961).

## 4. Discussion

Age, sex, employment status, and household size were found to be modest predictors of food waste behavior in Denmark, and age and employment status were found to be modest predictors of FWB in Spain. FWB in this study was measured by shopping routines, food preparation, and self-reported food waste, and thus, these socio-demographic factors predict the likelihood of respondents to buy and cook too much food, as well as throw out food. Similar to previous studies, socio-demographics only explained little of the variance of FWB in both countries [12,18]. Rather, attitudes, values, and other psychographic variables have been previously shown to be more closely related to behavioral outcomes [6,17]. Also, an individual's socio-demographic characteristics may not directly translate to household-level measures [37], limiting the predictive power of socio-demographics. Thus, one may conclude that socio-demographics play a small role in predicting household FWB.

Our results agree with exploratory studies that show age and the amount of food wasted to be negatively correlated [10–12,38]. Adults 65 years of age or older in particular have been found to practice food waste reducing behaviors such as planning meals in advance, and have more knowledge of food waste than younger adults [39,40]. In this study, the Spanish subpopulation of adults aged 65+ was very small, preventing the comparison of this age group with younger adults with regard to their FWB. Our results are also consistent with previous studies that have found a negative association between employment status and amount of food wasted [38,41], which have shown that compared to people working full-time, those not employed tend to waste less food.

Our study resulted in different associations between sex and household size and FWB for Denmark and Spain. While neither were significant predictors of FWB in Spain, they were both significant predictors of FWB in Denmark. The difference in results for household size may stem from the differences in household size between the countries, where the majority reported living in single or two-person households in Denmark, whereas the majority reported living in a household with three or more people in Spain. The present study found that living in a household with four people was associated with more FWB compared to those living in single-person households in Denmark. This finding may be attributed to household dynamics and cultural expectations. For example, those striving to be a 'good' food provider for the family have been found to have more food wasting behaviors in various contexts [14,42]. It has also been found that households with children are more likely to have more food wasting behavior [43], although in this report, household composition was not studied. Being male was associated with more FWB in Denmark, which is similar to findings in the United States and in the European Union (EU) [38,41], but different from Koivupuro et al., who found that males waste less food than females in Finland. This suggests that the relationship between sex and food waste may vary greatly between countries. The researchers in Finland speculated that women may be more likely than men to strive to provide their family with healthy, fresh products with a shorter shelf life, making them more susceptible to generating food waste [16]. However, Secondi et al. found that women on average in the EU appeared to be more conscious of food waste compared to men, which may make them less susceptible to generating food waste [38].

Our model of FWB, which combined food-related behaviors found to influence the amount of food wasted at the consumer level with self-reported food waste, was unique and provides a broader perspective on food waste. While the AVE of the latent variable FWB was just below the cut-off, the degree of convergent validity was similar to Mondejar-Jiménez et al.'s 'positive behavior toward food waste' (AVE = 0.497). As expected, self-reported food waste greatly underestimated the actual amount of food wasted in these countries. We found that 20% of the Danish and 40% of the Spanish

respondents reported to throw away *no* food in a regular week. When food waste was collected from a sample of Danish households and when food diaries were kept to record food waste in Spain, it was found that 97% of the sampled households generated food waste in Denmark (i.e., 3% of households produced no food waste) [22], and 80% in Spain (i.e., 20% of households produced no food waste) [44]. In addition to being prone to social desirability and memory bias [20], self-reported food waste is susceptible to underestimation due to an *individual* trying to assess *household* food waste [37]. As the self-reported food waste measure can bias individuals to underestimate the amount of food [20], combining self-reported food waste with other food-related behaviors was our attempt to minimize this bias. This approach is supported by previous research highlighting the complexity of food waste and arguing that food waste is not a single behavior, but rather a result of multiple food-related behaviors that lead to food being thrown away [10,13,39].

There are some strengths and limitations to this study. Strengths include a large sample size compared to previous studies [9,11,13,45] and the inclusion of respondents in two European countries. As partial measurement invariance was established, this study provides insight into country differences. A limitation to this study is that we measured individual behavior as a proxy of household behavior, as this approach has been shown to give an inaccurate representation of the entire household, especially when it comes to a habitual behavior such as food waste [37]. This may be another reason why socio-demographics are found to play a minor role in predicting FWB. Another limitation is the paucity of other possible predictors of FWB that have been shown in earlier studies to influence the likelihood or amount of food wasted. For instance, past research suggests that psychographic factors play a more important role than socio-demographic factors in explaining food waste at the consumer level [6]; however, our survey was designed for another research question, and thus was not designed to conduct a comprehensive study on the predictors of FWB. Also, socio-demographics such as income and household composition have been shown to influence food waste, but were not assessed in this study [46,47]. Furthermore, the data collection and sampling scheme limit the generalizability of the findings to the entire Danish and Spanish populations; thus, study findings should be interpreted taking into account the specific characteristics of the study samples.

Future research should approach food waste from a behavior perspective and conduct a more comprehensive investigation of predictors of food waste behavior. Survey questions measuring food-related behaviors should be pre-tested and piloted to ensure that there are no ambiguities in the questions and the respondents could understand the questions the way they are intended in order to prevent the deletion of items during measurement model assessment. Furthermore, studies should clearly differentiate between individual and household behavior when designing the survey. As it is difficult to measure household behaviors by only asking one individual per household, a suggestion would be to isolate individual food wasting behavior.

## 5. Conclusions

This study found that age, sex, employment status, and household size predict food waste behavior in Denmark and age and employment status in Spain. The findings support existing knowledge that socio-demographic factors are modest predictors of FWB. The present study contributes to food reduction strategies, as it gives a different perception of food waste incorporating a behavioral approach, and adds evidence from two European countries that socio-demographics should play a modest role in interventions. While food waste reduction strategies may benefit by targeting younger adults and those working full time in both countries—and men and individuals in bigger households in Denmark—strategies should be designed to target food-related behaviors that influence the amount of food wasted.

**Author Contributions:** All authors contributed substantively to the report. A.C.G. was the lead author on the manuscript and A.C.G. was responsible for the conceptualization in collaboration with C.v.D., I.A.B., M.R.O. and L.L. A.C.G. analyzed and interpreted the data with A.J.B. L.L. was responsible for the survey design and

implementation. Funding acquisition: I.A.B.; Supervision: M.R.O. and I.A.B. All the authors have approved the final manuscript.

**Funding:** Funding for this paper was provided by the European Union FP7 MooDFOOD Project 'Multi-country cOllaborative project on the rOle of Diet, FOod-related behaviour, and Obesity in the prevention of Depression' [grant agreement no.613598].

**Conflicts of Interest:** The authors declare no conflict of interest.

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
