# Peer review of "Socio-Demographic Predictors of Food Waste Behavior in Denmark and Spain"

_sustainability, doi:10.3390/su11123244_

Round 1

Reviewer 1 Report

I like this paper. It is great to see researchers move beyond TPB and develop a new model to try and explain this behaviour. Overall I don't think the current model advances our understanding of this behavior beyond previous research and this is fine. Even though the approach was different than TPB papers the amount of FWB explained by socio-demographics remains limited. I think this is an important finding actually. This may mean that it doesn't play a big role and/or there are issues with self-reported food wasting behavior. I think this needs to be better set up/elaborated in the in Introduction and Discussion.

Introduction:

Line 65-70 It is worth noting that socio demographic factors explain relatively small amounts of the variance of food wasting behaviour. See Visschers et al 2016 and van der Werf et al 2019 for examples.

van der Werf, P., Seabrook, J. A., & Gilliland, J. A. (2019). Food for naught: Using the theory of planned behaviour to better understand household food wasting behaviour. The Canadian Geographer / Le Géographe canadien, 63(2), 1-16. doi:10.1111/cag.12519

Line 69-70 While it is covered in the Discussion it would be useful to elaborate on the limitations of self-reported food waste in the introduction.

Methods:

Section 2.1

How many survey questions in the survey and how long did it take respondents to complete on average?

Are all the survey questions included in Table 1 and 2? If not suggest appendix that includes the full survey.

Table 1. How did respondent socio demographics compare to population socio demographics?

Results/Discussion:

Overall this study attempts to create a model of food wasting behavior and tries to move beyond using self-reported food wasting behavior as the exclusive measure of this behavior.

It seeks to provide an alternate approach/model to the theory of planned behavior, where as they have noted, the relationship between intention and behavior is often weak.

I find that 40% of Spanish respondents say they throw out no food waste unrealistic (even the 20% from Danish respondents seems low). This speaks to possible underestimation of self reported food wasting behavior. Part of this issue may be related to biases etc., part of it because people have highly variable assessment skills, and part of it because a single respondent is trying to assess household food wasting behavior. I wonder if for future research one would try to isolate individual food wasting behavior rather than group food wasting behavior. I would also move towards asking about frequency rather than trying to have respondents estimate a weight/volume.

Given the complex statistical analysis it would be useful to depict the final FWB model as a Figure.

It is interesting that the only AVE that is beyond the cutoff value of 0.5 is for self reported food waste. My takeaway from this is that it is not a very strong model. 

With regard to linear regression analysis the amount of variance explained by socio-demographic factors is very low and similar to similar studies by Visschers; van der Werf. The conclusion I draw from this is that using FWB model/index as a dependent variable does not result in a greater amount of variance being measured than these other studies. Why this is the case needs to be discussed in the Discussion. It may in fact be that socio-demographics play a small role in predicting food wasting behaviour. 

As a comparator it might be useful to develop two step linear regression models using self-reported food wasting as the dependent variable and then socio-demographic and then other behavioural factors (e.g., planning routines, shopping routines).

Alternately there may be other factors at play. In this research self-reported food wasting behavior is blended with other possible determinants. If self-reported food wasting behavior is suspect then this blending perhaps takes away from more meaningful factors.

Another issue is that an individual is not a household. See: 

Seebauer, S., Fleiß, J., & Schweighart, M. (2017). A household is not a person: Consistency of pro-environmental behavior in adult couples and the accuracy of proxy-reports. Environment and Behavior, 49(6), 603-637. 

This should be addressed in the Discussion section as a limitation.

Line 273, 327. I disagree with how socio-demographic factors are being framed. When they are only explaining 7.3/8.1% of the variance they are hardly 'important". They are at best modest predictors of FWB and as noted similar to other studies. I view this as a very interesting result. Is this because socio-demographic factors are only modest predictors at best or is it because potentially poor self-reported food wasting behaviours confounds the results?

Conclusions:

I disagree with the conclusions. The modest results are being over-stated. The real conclusion is that socio-demographic factors appear to play a modest role and that they should play a modest role in interventions. The conclusions should include some next research steps. How do we get past issues such as 'an individual is not a household', self-reported food wasting behavior etc.

Author Response

Dear Reviewer,

Thank you for your comments on our manuscript entitled ‘Socio-demographic predictors of food waste behavior in Denmark and Spain.’ We found the comments to be very helpful and to greatly improve the quality of our manuscript. We have addressed the issues pointed out and changed the manuscript accordingly.

Our point-by-point responses to your comments follow in the attachment. All authors have read and approved the revised manuscript.

Sincerely,

On behalf of all authors of this manuscript,

Alessandra Grasso

Reviewer 2 Report

The proposed research «Socio-demographic predictors of food waste behavior in Denmark and Spain » falls within the scope of Sustainability. According to the reviewer’s opinion, minor revisions are required in order to accept this research study for publication in Sustainability.Please, comply with the following suggestions and comments: Comment 1: The paper is in general well accompanied of clear explanations.However, questions that need to be answered: Why your study is important? How it extend the existing knowledge on the issue/topic? Concluding remarks – authors must elaborate more on what is their contribution to the literature. Comment 2: More recent papers in the field should be integrated in the literature review. Comment 3: Finally, when you submit the corrected version, please do check thoroughly, in order to avoid grammar, syntax or structure/presentation flaws - please seek for professional English proofreading services or ask a native English-speaking colleague of yours in order to refine and improve the English in your paper.

Author Response

Dear Reviewer,

Thank you for your comments on our manuscript entitled ‘Socio-demographic predictors of food waste behavior in Denmark and Spain.’ We found the comments to be very helpful and to greatly improve the quality of our manuscript. We have addressed the issues pointed out and changed the manuscript accordingly.

Our point-by-point responses to your comments follow below. All authors have read and approved the revised manuscript.

Sincerely,

On behalf of all authors of this manuscript,

Alessandra Grasso

Reviewer 3 Report

Thank you for submitting your manuscript to the Sustainability journal. The manuscript respected the formal requirements in terms of layout and in terms of strcuture (Scientific Best Practice).

The introduction is poor. In the introduction, you need to connect the state of the art to your manuscript goals. Please follow the literature review by a clear and concise state of the art analysis. This should clearly show the knowledge gaps identified and link them to your manuscript goals. Please reason both the novelty and the relevance of your manuscript goals. In your case. this means to provide figures about the quantities of food waste in Europe and the respective countries and to connect them with the need to reduce food waste. Further, yu should make reference to sustainable consumption.

In the materials and methods section you need to explain the criteria for the selection of the questions /topics put to the interviewed persons to make clear to the reader what you intended with the question.

The conclusions are by far too short.

Author Response

(The authors gave the same response as above.)

Round 2

Reviewer 1 Report

Thank-you for addressing my comments. I think it makes for a stronger paper.